# Job Demands, Resources and Strains of Outpatient Caregivers during the COVID-19 Pandemic in Germany: A Qualitative Study

**DOI:** 10.3390/ijerph18073684

**Published:** 2021-04-01

**Authors:** Natascha Mojtahedzadeh, Tanja Wirth, Albert Nienhaus, Volker Harth, Stefanie Mache

**Affiliations:** 1Institute for Occupational and Maritime Medicine (ZfAM), University Medical Centre Hamburg-Eppendorf (UKE), Seewartenstr. 10, 20459 Hamburg, Germany; n.mojtahedzadeh@uke.de (N.M.); t.wirth.ext@uke.de (T.W.); harth@uke.de (V.H.); 2Department of Occupational Medicine, Hazardous Substances and Public Health, Institution for Statutory Accident Insurance and Prevention in the Health and Welfare Services (BGW), Pappelallee 33/35/37, 22089 Hamburg, Germany; a.nienhaus@uke.de; 3Institute for Health Service Research in Dermatology and Nursing (IVDP), Competence Centre for Epidemiology and Health Services Research for Healthcare Professionals (CVcare), University Medical Centre Hamburg-Eppendorf (UKE), Martinistr. 52, 20246 Hamburg, Germany

**Keywords:** outpatient care, COVID-19, occupational demands, occupational resources, strains, health and safety

## Abstract

The COVID-19 pandemic has affected health professionals in a special way, as they are responsible for the care of vulnerable groups. Little is known about how outpatient caregivers perceive their working conditions during the pandemic in Germany and about the difficulties they face. The aims of this study were (1) to examine specific job demands of outpatient caregivers in regard to the COVID-19 pandemic, (2) to illuminate their job resources they can rely on and (3) to identify potential strain reactions they experience. Fifteen semi-structured telephone interviews were conducted with outpatient caregivers working in Northern Germany in the period May–June 2020. Interviews were analyzed by using qualitative content analysis. Outpatient caregivers experienced daily mask obligation, lack of personal protection equipment (PPE) and stricter hygiene regulations as demanding during the pandemic. They also described a higher workload and emotional demands such as fear of infection or infecting others. They perceived team spirit and communication as important work-related resources. Depressive symptoms and feelings of stress were described as strain reactions. Outpatient care services need to be better prepared for sudden pandemic situations and provide their employees with sufficient PPE and education to reduce pandemic-related job demands leading to negative strain reactions.

## 1. Introduction

“Epidemics are health emergencies in which human life is threatened and there are significant numbers of sick and dead” [1]. In respect of the current outbreak of the new type of coronavirus (SARS-CoV-2) and its globally rapid as well as dynamic distribution, there is also a constantly increasing number of infected people in Germany [2]. COVID-19 can especially be dangerous for vulnerable person groups, such as the elderly [3] and people with previous basic diseases like coronary diseases, diabetes, diseases of the respiratory system, liver and kidney [4]. “Because the virus is transmitted from human to human, those in jobs requiring contact with other people are at a higher risk of contracting it.” [5]. In 2019 there were 14,688 outpatient care services in Germany [6] with 421,550 employees who were working in outpatient care [7]. Most of them had professional qualifications such as geriatric nurses (98,976), geriatric care assistants (21,831), health and care nurses (78,129) or nursing assistants (14,822) [7]. Work activities in the outpatient care include body-related care measures (personal hygiene, nutrition, promotion of mobility), nursing care measures (e.g., help with orientation, organizing everyday life or maintaining social contacts) and home nursing (administration of medication, bandage changes, injections) [8]. Outpatient caregivers are; therefore, bound to their work in the field service and to the direct contact with patients [9]. Although there was a decided contact ban by the German federal government [10], outpatient caregivers are forced to visit their clients anyway and carry out their working activity on the patient [9]. Having to work in the outside field, there is a high work-side risk for COVID-19 for outpatient caregivers themselves [1]. Until the 5 June 2020, there were 14,120 suspicious cases of coronavirus infections in Germany among healthcare and social workers (reported to the Institution for Statutory Accident Insurance and Prevention in the Health and Welfare Services (BGW)). Of 4850 cases with a known positive test result, there were 1688 positive cases in inpatient and outpatient care in total and five have even died [11]. Risk of people getting sick or dying due to pandemics can also cause psychosocial strain [1]. Outpatient caregivers’ patients are people in need who are frequently over 60 years old [12]. Hence an infection could be problematic, since the elderly are defined as a vulnerable group because of their age and possible previous diseases [3]. Furthermore, perceived stress levels can be increased in times of epidemics [1]. Therefore, it can be assumed that outpatient caregivers are exposed to particular demands during the COVID-19 pandemic. It is already known that the coronavirus pandemic has a negative impact on healthcare workers such as physicians, nurses, and auxiliary staff. Especially post-traumatic stress and symptoms of depression can appear among healthcare employees during the pandemic, as well as anxiety and insomnia [13].

### 1.1. Theoretical Model

The Job Demands-Resources Model (JD-R Model) by Bakker and Demerouti [14] served as a theoretical framework to explain job demands and resources in the outpatient care in our qualitative study. According to the concept of the JD-R-Model, there are specific job demands as well as resources which need to be considered in each work activity. Job demands can be divided into physical, psychological, social and organizational factors, such as a high workload, a negative influencing work environment as well as emotional demanding interactions with fellow human beings. Job resources can be the achievement of personal goals, personal development as well as autonomy and support, whereby resources can be influenced by the individual’s motivation to work [14]. Job resources can “buffer” job-related demands [15]. An imbalance of job demands (time pressure, work load, etc.) and job resources (rewards, support, etc.) can lead, in the long run, to negative strain reactions [15], such as burnout [16]. There are two psychological processes relevant for developing strain reactions and motivation. On the one hand, employees’ psychological and physical resources can be overstretched and lead to stress and exhaustion [14]. This is called the “health impairment process” [14]. On the other hand, job resources can have a motivational nature and; therefore, be influenced by the individual’s work engagement [14].

### 1.2. Current State of Research

#### 1.2.1. Job Demands and Resources in the Outpatient Care

Previous studies have shown that outpatient caregivers are confronted with several demands in their daily work. Besides hiding their own emotions, such as anxiety and fears of death and pain [17], moods and feelings can also have a direct impact on caregivers themselves [18]. This requires constant adaption and might be very demanding [19]. Further demands could be a poor cooperation between caregivers and patients, which can lead to complications in their relationship and care activity [20]. They, furthermore, need to abide by the rules of the homeowners [21], which can lead to personally unpleasant situations [9,22]. Another requirement is that the caregivers’ work equipment must be brought to the patients [21]. Having to fill in due to colleagues’ sickness, lack of communication and support are also demanding factors in outpatient care [9,21,23]. High work demands and work density can promote stress reactions and time pressure [23,24,25,26]. Stress experience is common in the outpatient care [27]. Work interruptions resulting from traffic, bad weather conditions and staying longer than expected with patients can be listed in this context [9,22,23]. It has been shown that, in some cases, rest breaks cannot be taken due to excessive work tasks and regeneration in terms of days off are not planned wisely [9,24]. Working by themselves can cause feelings of social isolation and, beyond that, feedback from superiors might be missing [9,22,23]. In addition, sexual assaults by male patients towards female caregivers are reported [9,22]. High personal responsibility, self-determination, independent working and participation opportunity are described as job resources in the outpatient care [9,28]. Team meetings give them the possibility to discuss cases as well as the chance to reflect their work shifts together. Feeling satisfaction and meaningfulness regarding their occupation is an important resource [21] as well as social support by colleagues and employers [29].

#### 1.2.2. Job Demands, Resources and Strain Reactions in the Course of the COVID-19 Pandemic

There are already studies that have analyzed the demands of healthcare professionals and/or the impact on mental health during the COVID-19 pandemic [30,31,32,33,34,35,36,37,38,39,40,41,42,43,44,45,46,47]. Most of the existing studies examined medical staff from China (cf. short current review by [48]). The studies often included healthcare workers. Among them, inpatient-nursing staff were also surveyed [30,31,32,33,35,36,38,39,44,47]. A report of a German study, which analyzed partly inpatient as well as outpatient care services, illustrates specific work-related demands during the COVID-19 pandemic. These extend from fear of infection, absence due to illness of colleagues, shortage of personal protection equipment (PPE) as well as disinfectants and financial insecurities up to a higher workload since the start of the outbreak of the coronavirus [49]. Stress, anxiety, depressive symptoms and/or further negative impacts on healthcare workers’ mental health were described in these studies, whereby nurses were more affected than other healthcare workers, such as physicians (cf. [43]). Li et al. [34] focused, in their descriptive online survey, on frontline nurses (*n* = 234), nursing staff (*n* = 292) and the general population (*n* = 214) in China. Traumatization due to the coronavirus pandemic existed in every group of them. Mo et al. [37] analyzed stationary nursing staff from China (*n* = 180): fear, high workload and having kids caused stress. Shen et al. [40] surveyed 85 ICU nurses from China. The participants reported on their limited experience with a pandemic, higher workload and fear of infecting their family members. Furthermore, a qualitative study on 23 frontline nurses in China added to their fear of infection, lack of knowledge about the coronavirus, fear of death, work pressure, isolation/loneliness and lack of PPE. This led to depressive symptoms, anxiety and fear [46]. There is one online survey analyzing German medical professionals (*n* = 2827; 65.6% doctors, 29.5% nursing staff, 4.9% other). Nurses reported significantly higher stress than doctors, while doctors were more concerned about their own health than nurses (43.5% vs. 38.6%) [45]. Another study at a German university hospital (*n* = 35 doctors, *n* = 75 nursing staff) has focused on the comparison of psychosocial strain between physicians and nurses. Especially nurses of the “corona station” reported higher stress and exhaustion levels, depressive symptoms as well as less occupational fulfilment. Uncertainty about the pandemic as well as risk of infection were factors causing these negative strain reactions [43]. Risk of infecting themselves or patients was stressing for nursing staff in quarantine in a Korean children’s hospital [41]. A Brazilian cross-sectional study examined nursing professionals at a university hospital. Of 88 nurses, 48.9% reported anxiety and 25% depressive symptoms; however, job demands were not described [42].

The current further spreading COVID-19 pandemic is without a question burdening for caregivers. The studies described above show a negative impact on healthcare workers’ (stationary setting) mental health (e.g., stress, anxiety and depressive symptoms). However, concrete demanding factors are seldom described and analyzed. The focus of the majority of the studies was more on the consequences, except for Zerbini et al. [43] and for Wolf-Ostermann et al. [49]. Job resources of nurses in times of the COVID-19 pandemic are described less frequently. Teamwork, gratefulness of and appreciation by others, as well as social support in work as well as in private life and also free time are the most helpful resources [39,43,44]. Feeling proud of their job can make coping with challenges of the pandemic easier for frontline nurses [46]. Most of these studies are in reference to stationary care or to frontline nurses except for the study by Wolf-Ostermann et al. [49]. Job resources considering pandemic situations for outpatient caregivers in Germany have not been examined yet.

In conclusion, the primary focus in the scientific literature has been directed to healthcare workers or respectively to hospital nursing staff. Studies analyzing job demands and resources of German outpatient caregivers during the coronavirus pandemic are missing largely. The occupational health of these employees is; however, highly relevant with regard to the increasing number of people in need of care in Germany who use outpatient care services [12]. Particularly considering the fact that they are still forced to go outside and expose themselves to a possible risk of infection [9]. To obtain the health of outpatient caregivers, a concept of occupational health and safety is relevant which requires prior identification of possible demanding factors (e.g., via a risk assessment which is mandatory for all employers in Germany) [50,51].

### 1.3. Study Aims and Research Questions

The aim of this study was to examine the job demands and resources as well as strain reactions of outpatient caregivers during the coronavirus pandemic in Germany. In adherence to Bakker and Demerouti [14], we focused on various demanding factors and resources of the outpatient care occupation. 

We proposed the following research questions:What are the specific job demands of outpatient caregivers with regard to the COVID-19 pandemic?What are job resources outpatient caregivers can rely on while working during the COVID-19 pandemic?Which individual strain reactions do outpatient caregivers perceive?

## 2. Materials and Methods

### 2.1. Methodological Orientation and Theory

Using qualitative explorative research methods [52] as an underlying theoretical framework, outpatient caregivers’ job demands and resources during the COVID-19 pandemic should be explained. Since job demands and resources of outpatient caregivers in Germany during a pandemic have hardly been studied before, it is quite reasonable to use an explorative research design by conducting interviews [53,54]. Open conversations focus more on personal experiences rather than on direct answers to the original research question, which was an aim of this study [55].

### 2.2. Study Design

The present study followed a qualitative research approach, having a deductive-inductive procedure [52]. Opinions and experiences of the outpatient caregivers interviewed were the focus of this study [56], “subjective truth and social sense structures” should be reconstructed [57]. Prior knowledge through literature research builds the basis of the asked questions in the interview [58]. Hence semi-structured interviews were chosen. Because of the COVID-19 pandemic it was necessary to keep personal distance whenever possible [2]. Hence personal interviews with outpatient caregivers were not feasible. Eventually the study group decided to do telephone interviews.

### 2.3. Participant Selection and Interview Conduct

We conducted 15 semi-structured telephone interviews with outpatient caregivers from outpatient care services in Hamburg, Germany. Nine interviews were conducted in May 2020, the remaining six were carried out in June 2020. The interviews were conducted by one female research associate (NM) working in the field of “occupational health psychology”. A purposeful sampling was applied. Interviewees who were working as an outpatient caregiver for at least six months, working in Hamburg, Germany and who were fluent in the German language were eligible and have been recruited. No specific profession in outpatient care was required (cf. [8]). Outpatient care services were contacted via invitation emails and telephone calls by the interviewer herself. Invitations were also made online on social media. Study participation was voluntary. Prior to the interviews, participants were asked to sign a declaration of informed consent regarding the performance and recording of the interview. All participants were in a position to understand and consent to the study requirements and provided written informed consent. The interviews were conducted until no new topics were identified (i.e., data saturation was reached). The language in all interviews was eventually German. All telephone interviews were tape recorded. Interview length was from 26 up to about 60 min. Participants were told that they were able to terminate the interviews at any time. No non-participants were present during the interviews. No repeat interviews were carried out. Field notes were made immediately after each interview.

### 2.4. Interview Guideline

A semi-structured interview guideline was designed within the general framework of the empirical and theoretical background. At first, adequate questions were collected. Afterwards, questions were reviewed and sorted. Finally, questions were subsumed in categories [57,59]. In adherence to Misoch [60], the structure of the interview guideline was then divided in four phases: the information phase, the warm-up phase, the main phase and, finally, the final phase or rather the end of the interview. An extract of the interview guideline is shown in Table 1. The guideline included further questions on coping strategies, support needs, health behavior, health promotion, occupational health and safety as well as further requests, which will be presented elsewhere. A pre-test interview was performed before the actual first interview in order to receive feedback from research colleagues and to improve the interview guideline where applicable.

### 2.5. Analysis

All audio recordings were transcribed verbatim following Kuckartz [58]. Afterwards, the transcripts were anonymized and analyzed in a deductive-inductive process according to the qualitative content analysis of Mayring [61]. We used MAXQDA 2020 (VERBI Software 2019, VERBI GmbH, Berlin, Germany) for data analysis [62]. In an iterative process, the main researcher identified and refined codes, categories and sub-categories. Coding was reviewed reciprocally for accuracy and was carefully debated with another researcher until consensus in terms of the final coding system was achieved. The final coding system was summarized in another separate document in which the material was further diminished and compressed by two members of the research team. During the process of analysis, reflexivity and transparency relating to the potential influence of the researchers’ objectives and prejudices on the results, as well as interpretations, were constantly emboldened. Transcripts and results were not returned to the interviewed person although they had the possibility to claim them. All quotes used for publication purposes were translated from German to English. 

## 3. Results

### 3.1. Sample Characteristics

As depicted in Table 2, interviewees were 21 to 67 years old. Of the 15 outpatient caregivers from Hamburg, Germany, three were male and 13 worked full-time with a work experience range from seven months up to 36 years, so all of them had been working in the outpatient care for at least six months. Most of the 15 interviewees were qualified as geriatric nurses. Additionally, six of the caregivers had at least one child at home to whom they were responsible.

### 3.2. Job Demands during the COVID-19 Pandemic

From the interviews the following six main categories relating to job demands were identified: work organization, work task, quantitative demands, work environment and tools, social relationships, and emotional demands.

#### 3.2.1. Work Organization

According to outpatient caregivers, work organization structures have changed since the beginning of the outbreak of the COVID-19 pandemic. Considering the responses of the interviewees, three subcategories have been transpired next to general organizational advices: handing over reports during the COVID-19 pandemic, working time and filling in for colleagues. Challenges occurred for instance in handing over reports to colleagues which had to be carried out in written form instead of in a personal conversation. So there was a potential higher risk for misunderstandings between colleagues.


*“(…) the topic of handovers is really important in the outpatient care sector. (…) but there is no exchange of information from face-to-face anymore, mainly via documentations in written form. (…) Since you have to be very independent in the outpatient care regarding medication procurement or prescription, there is the possibility of information unwillingly getting lost somewhere which will lead to unnecessary stress the next day. That is difficult.” (Interviewee #7)*


Moreover, working time of interviewed outpatient caregivers varied. Some interviewees reported no change in their working time since the COVID-19 outbreak. The fact of a high working time before the pandemic was also mentioned. Still, many outpatient caregivers stressed a higher working time due to the pandemic, especially at the beginning of the outbreak where there were times they had more work to do altogether.


*“There is more work. (…). Especially at the beginning, when it was starting and we were getting more orders. (…)”. (Interviewee #2)*


Filling in for colleagues who had to stay home due to quarantine regulations or for sick staff was stressed by outpatient caregivers as a negative impact of the pandemic on their work situation. Older employees had the impression that younger staff were making use of that quarantine regulation on purpose.


*“I often have to fill in for constantly sick colleagues and there was also the quarantine regulation for people coming back from risk areas who had to stay home for 14 days. That was often occupied by younger colleagues.” (Interviewee #14)*


Furthermore, outpatient caregivers found themselves looking at several challenges in relation to their general conditions. For example, the obligation to accept patients coming from a hospital which reportedly had several COVID-19 patients, was perceived as harmful as they could carry the coronavirus into the care service. Reorganizations regarding the work organization were also described as demanding by outpatient caregivers. A displacement of internal rooms, a postponement of induction training as well as caregivers’ no longer switching in the treatment of patients were stated in this context.

#### 3.2.2. Work Task

Regarding their work tasks, outpatient caregivers described, on one hand, their self-reliance and, on the other hand, the process of handling possible coronavirus positive patients as demanding. The perceived increased responsibility and self-reliance due to the pandemic was another psychological demanding factor. Besides focusing on their responsibility towards patients, they experienced a higher self-reliance in relation to the execution of correct working steps, particularly with regard to the intensified hygienic requirements due to the pandemic.


*“Well, it is kind of difficult to describe (…), but my head is constantly rattling all the steps through, so now each step is reconsidered twice or for three times extra.” (Interviewee #7)*


Some interviewees stressed an in-house made pandemic plan that has educated them about each step in the process of working during the COVID-19 pandemic. However, the majority complained about the lack of standardized processes, let alone a specially developed plan. For instance, in case of a coronavirus positive tested patient, interviewees’ descriptions of processes were different. Many outpatient caregivers reported that patients would be further treated in consultation with their general practitioner by making use of special PPE. They also described reporting a COVID-19 patient to the local public health department, pulling out the outpatient caregiver who took care of that patient and contact tracing as paramount. Having no knowledge of the needed procedure and the lack of unified processes of what to do exactly overall caused feelings of insecurity due to missing information.


*“The procedure in our outpatient care service would make me skip any work shift and stay home until my test result would come in. All clients would be treated by the head of the care service while putting on personal protective equipment only. (…) The course of disease will decide whether the patient needs to get to the hospital or whether outpatient care is still possible or not.” (Interviewee #11)*


#### 3.2.3. Quantitative Demands

Higher quantitative demands have evolved by new duties due to the COVID-19 pandemic causing additional work and time pressure in general. New strategies in terms of infection control had to be implemented, for example permanent hand hygiene or disinfecting the cars or used objects. Strengthening awareness of reinforced hygiene routines was also mentioned in this context. Increased hand washing and disinfecting that was more necessary than before was listed as impairing the skin which was perceived as demanding.


*“(…) I’m always in these gloves while working so I feel like I’m more exposed to disinfectants and my fingers are feeling like that already.” (Interviewee #2)*



*“We have to pay even more attention than usual to hygiene procedures. We are requested to do daily fever measurements and to document every single symptom (…).” (Interviewee #8)*


Moreover, government-mandated daily fever measurements were experienced as incriminating by many outpatient caregivers as it was described as annoying and leading to a higher workload.


*“Well, we have received the regulatory order to measure temperatures of the patients daily. I find that very annoying.” (Interviewee #11)*


Beyond this, it was time-consuming to obtain thermometers after receiving the regulatory order requesting to measure each patient’s temperature daily.


*“(…) and then all colleagues were driving around for three days, checking every store to get thermometers. You can’t give an employee only one thermometer which they take with them to all the patients. So we need to have one individual thermometer for each client. It was an insane action to get those.” (Interviewee #4)*


Some of the outpatient caregivers have complained about time pressure during their work since the beginning of the pandemic. For that, different reasons were responsible, such as the stay in practices or pharmacies and supermarkets which was contingent on distance regulations, as well as traffic and the fact that outpatient caregivers were not given the possibility to use patients’ restrooms anymore. Every time they needed to go to the restroom they had to return to their workplace again.


*“For instance we have to get a prescription from a practice or we have to get medications from a pharmacy. Because of the coronavirus we have to wait—for example in the practice there were only two person allowed in there. So we have to wait for like half an hour outside until the other people are finished. And in the pharmacy we have a waiting period, too. Every ten minutes someone is allowed to go in only, while wearing masks and so on.; therefore, there is delay everywhere (…).” (Interviewee #10)*


The beginning of the pandemic led to cancellations on the part of patients of outpatient care services. So somehow an underchallenge was perceived by a few outpatient caregivers. However, a high flood of information as well as other obligations towards patients, such as flower watering or grocery shopping, resulted in more work or, rather, an overload for the outpatient caregivers.

#### 3.2.4. Work Environment and Tools

Outpatient caregivers’ interview answers revealed, on one hand, demands in their daily work environment. On the other hand, there were demanding factors highlighted regarding their work tools. Outpatient caregivers pointed out that it might be quite impossible to keep the needed distance between humans during their work activity which caused certain discomfort and fear in general.


*“(…) it’s not always possible to keep the distance to patients, for instance when showering, applying lotion or putting on their clothes, you cannot keep the distance.” (Interviewee #5)*


The work environment of outpatient caregivers has been classified to be in the field service so there is always a chance to infect themselves as well, especially because some of them had to use public transportation to get to the patients. Individual perception of personal risk of infection during work varied from not at all up to an increased risk. Some highlighted the fact that having contact with other people and working in the field service might increase their risk of getting infected in comparison to employees working from home. Others perceived their risk as low due to good hygiene measures.


*“Well, there is always a risk. But I’m doing everything that I can to not get infected.” (Interviewee #5)*



*“Quite low. Because we have all materials, I think we even might be more protected than the general citizens. Since we have hand disinfectants, enough masks and enough gowns. And more important: we know how to use them.” (Interviewee #8)*


Regarding work tools, mask obligation at work was frequently mentioned as a demanding factor. During their work shifts, there were times in which the outpatient caregivers had trouble breathing because they were obligated to put on a mouth-nose protection mask while treating patients all the time. It was too hot, especially when showering patients. Hampered communication to other people and patients as well as feelings of pain in the lungs area were stated by some caregivers as well. Overall, all outpatient caregivers felt that wearing a mask during work, except when driving, was stressful.


*“(…) when I’m at work I have to keep the mask on the whole time, except for the times I’m driving in my car. I feel I’m not able to receive enough air to breathe.” (Interviewee #2)*



*“(…) it’s of course annoying at work. Especially when we have to do something, I don’t know, have to shower someone, it’s warm and stuffy and one has a silly mask on. Well that is just annoying (...).” (Interviewee #4)*


Yet, some outpatient caregivers doubted the safety provided by wearing the masks. Reasons were, on the one hand, available masks lacking in sufficient quality. Special masks would be needed for better protection. On the other hand, the fact that the masks are not changed after each patient would unsettle the outpatient nursing staff. 


*“It’s of course really difficult looking at the coronavirus situation because there are a lot of gaps, it starts with the masks. This is a great gap. In theory you have to use a new mask after each patient, it doesn’t matter whether it’s fabric or disposable, just to cut off the line there a little bit more.” (Interviewee #12)*


Many outpatient caregivers experienced a lack of PPE, especially at the beginning of the COVID-19 pandemic. Since the procurement of PPE was challenging, missing gowns, protections masks as well as disinfectants were described as burdening by interviewees.


*“(…) it was burdening that personal protection equipment was not available, such as disinfectants, FFP-2-masks and so on, that was really tiring then. I was afraid, how could I protect myself? (…).“ (Interviewee #6)*


#### 3.2.5. Social Relationships

Outpatient caregivers described demands considering their social relationships with patients, patients’ relatives, their colleagues and their executives. They came across several challenges in relation to the treatment of patients. Patients’ insecurities concerned especially the whole pandemic situation in general. Handling their fears and worries were described as exhausting by a lot of outpatient caregivers during the COVID-19 pandemic. Moreover, patients were scared to get infected by the outpatient caregivers themselves. Mimics were also lost due to the masks which caused a more complicated communication between outpatient caregivers and patients which in turn resulted in more anxiety of the patients. That again was perceived as tiring by the respondents.


*“You would want to care face-to-face because there is usually an emotional connection to clients, no matter if they can communicate or not, you look each other in the eyes, you talk to each other and then the mask, it is somehow not the same.” (Interviewee #7)*



*“Well, a special challenge is of course handling the patients’ fears or to handle everything in the environment, that is very, very difficult. To convince patients that we are experts and that we know how to protect ourselves and to protect our patients of course and that we don’t expose them to something that they could get sick by. And this conviction takes really a lot of energy and they are so fearful and so on (…).” (Interviewee #9)*


Building and maintaining relationships with patients has proven to be more difficult overall. Conversations with patients, for example, were perceived as challenging for many outpatient caregivers. One topic often mentioned was the enlightenment of patients about the pandemic situation in general. They also frequently highlighted conversations to calm down patients, which was characterized as strenuous. Talking with patients to reduce their aloneness as well as to explain why masks are being worn were also named by some of the interviewees.

Observed destructive behavior of patients was also worrying outpatient caregivers. Patients showing aggressive as well as sad or scared behavior patterns were felt to be burdensome for some of the interviewees. Furthermore, scaremongering by disinfecting after outpatient caregivers themselves was perceived as negative. Above this, cancelling care was criticized by the care experts as some patients were in need of professional wound care.


*“(…) so we had a very difficult case, she really disinfected after us after every move. That was very awful.” (Interviewee #8)*



*“I had to calm down one patient who was scared of death. She was also dement which was difficult in combination with the fear of death, she was aggressive. But she was bedridden so I wasn’t scared that she could harm me but of course, I had to stay there and be patient and careful, step-by-step I needed to reduce her nervousness.” (Interviewee #10)*


Outpatient caregivers have also mentioned new challenges regarding the interaction with patients’ relatives. Relatives were insecure and had a negative attitude towards outpatient caregivers. Consequently, conversations with them needed much more empathy and comprehension according to respondents. The education of relatives about the pandemic was also mentioned.

*“And then the conversion of relatives suddenly being very mad. One has to spend many hours to explain to relatives which measures are needed and that it is not possible to let them enter the room, even if they are treated ambulatory in the treated environment and they don’t get it. And then they try to come in anyways and yes, many are claiming that caregivers have to think of all the interventions themselves and try to bypass all the recommendations.” (Interviewee #13*)

Distrust towards colleagues concerning their compliance with respect to social distancing was mentioned a couple of times. Worries especially referred to younger colleagues who might be more reckless regarding hygiene regulations and social distancing. 


*“What will happen in the end, you don’t know, I don’t think that anyone will stick to it 100 percent, it’s just like that.” (Interviewee #2)*


A lacking communication with executives about fears and worries was also reported at some point which was perceived as improvable.

#### 3.2.6. Emotional Demands

Outpatient caregivers were confronted with several emotional challenges while working during the coronavirus pandemic, which included difficulties in not only their working life, but also private complications which were negatively impacting their work. For one thing they had to hide their own feelings in front of other human beings, especially when it came to patients. Reasons were not only to spare patients but also management instructions not to show their emotions in front of the patients. In addition, they felt emotionally stretched because of the permanently thematized topic of the pandemic. Furthermore, patients’ sadness up to breakdown, particularly also the fact that dying patients weren’t allowed to welcome visitors, were emotionally demanding.

*“Well, one always looks whether it is an adequate situation or not. So when I’m with an already crying client because they are scared, of course I won’t say “I’m also scared you might get it”. One is clearly aware of that.” (Interviewee #7*)

A further challenging factor in this context is discrimination, which was stressed by the interviewees. Outpatient caregivers talked about being discriminated by patients themselves. This was made noticeable by disinfecting behind care experts or not being able to use patients’ restrooms anymore. Moreover, some outpatient caregivers reported going into patients’ homes after being asked whether they were healthy or not or having been requested to wrap themselves up completely before going in. There was also discrimination by patients’ relatives experienced, such as clear requests not to touch anything or to disinfect after the care service. Being forced by governmental regulatory to measure patients’ temperature daily was found to be some kind of discrimination as well as the fact of hospitals suddenly refusing to take trainees for their needed internship. Moreover, outpatient caregivers perceived prejudices referring to their profession by the general public. They were seen as a potential risk factor, respectively transmitter of the coronavirus due to their job as an outpatient caregiver.


*“And also, when one has sneezed or something like that, it caused sudden panic: ‘Are you healthy? Do you have something?’ Doors were opened and ‘Do you have a cold, are you coughing?’ (…). And when people know what you are doing professionally, like my neighbors in my building, they did fear me. And if I’m thinking about it, my son is living with me, his father clearly stated: ‘You know what your mother is doing professionally, I would like to see you but I’m scared that you will drag in something in my house’.” (Interviewee #11)*


Another demand for some of the outpatient caregivers was the fear of getting infected themselves during the pandemic. Their profession in general was a favoring factor for that fear. 


*“That’s indeed always there.” (Interviewee #13)*


In addition, many outpatient caregivers emphasized their constant fear of infecting other human beings, especially their own family members. Being afraid to be a transmitter of the coronavirus in general and then infect patients were also described as huge demands. 


*“(…) I’m rather scared for my relatives, the clients or colleagues. Me myself, I’m 25, I’m completely/ I’m not afraid.” (Interviewee #7)*



*“An insanely big fear of mine is to infect clients. Despite of hygiene measures. And if there was a difficult fatal course of events, I think, I don’t know what would be. In the beginning, I even thought or rather I still think that I would need powerful psychological support, because I then would have someone on my conscience. And that’s the opposite of one’s expectations and one’s profession.” (Interviewee #11)*


Ultimately, colleagues’ fear was tiring on top for a few outpatient caregivers who had to deal with their own emotions.

The COVID-19 pandemic as a consequence also seems to have an impact on the personal lives of outpatient caregivers. The following describes several effects on personal lives which outpatient caregivers had to handle throughout their daily work as well.

Many outpatient caregivers suffered from social isolation due to the request of social distancing. The need of social contacts could not be satisfied as outpatient caregivers were not able to see their relatives or to join social events because they were not permitted at that time. Interpersonal relation felt more distant than before, since the use of digital media was perceived as less satisfying for the outpatient caregivers.


*“I’m not that happy anymore because I am not able to create my leisure time as I’m used to. I’m a person who likes alone time but they took my choice to go for like shopping or to see other people. To me that’s really awful, to me that’s kind of social isolation which is really hard to deal with and that make sometimes mad (…).” (Interviewee #9)*


Some told about financial insecurities which were caused by the lacking use of ambulatory care services. As a result of patients cancelling their care appointments, order quantities have decreased, which led to worries on the financial side.


*“(…). Or that one client suddenly called and said: “Well, I won’t be there anymore. I’ll return when all of this is over.” (…). And that’s difficult because there is no security and financial worries can emerge.” (Interviewee #13)*


The outbreak of the coronavirus also caused further challenges regarding outpatient caregivers’ personal lives. Many interviewees were confronted with having to take more care of their child/children. This partially meant home-schooling as well as care for their kids. Outpatient caregivers described this as a double burden which was not only tiring after work, but also caused a lack of sufficient recovery. Moreover, disputes with children or partners have increased since the pandemic. However, not being able to see their kids in order to protect them from getting infected caused a bad conscience. Feeling like there was not enough private social support and having to educate people in the private environment were also reported. 


*“One has to be caregiver, mother and teacher all at the same time, that’s just like that (…).” (Interviewee #1)*



*“Well, to me one big challenge is my family. I have massive fights with my son who is chronically underchallenged, who has arguments with me and who bumps with me, which he usually gets done in the schoolyard. Now he is taking care of that with me at home.” (Interviewee #12)*


Worries about their children as being demanding was additionally reported by outpatient caregivers who were parents. On this occasion, the fear for their children getting infected was in the foreground, also for them being in the transmitter position and a threat to their patients. Of the further, they were afraid of their children becoming isolated due to closure of schools.


*“(…) Because of my son. Because he’s not able to go to school and you fear that he could grow lonely completely.” (Interviewee #3)*


The uncertainty about consequences of the pandemic, were also perceived as threatening and thus overstraining.


*“The overload is the unknown. And how long. How long will that be? How long one can buffer certain things.” (Interviewee #11)*


### 3.3. Work-Related Resources

In contrast to demanding factors caregivers of the outpatient care had to face every day since the outbreak of the coronavirus, work-related resources have also been described during the interviews. In general, a few have talked about an open culture in their care services where critics and suggestions were possible. A general satisfaction with work, its premises, feelings of security, for instance by the possibility to go by bicycle, as well as the social interaction with patients were perceived as resources at work. Hygiene measures and professional know-how were seen as protective factors, as well as the fact of working alone and driving by car. In addition, a time saving due to less traffic at the beginning of the pandemic, existing PPE, the possibility to keep the distance sporadically and the consideration of childcare were also mentioned. Meetings where possible and harmonious relationship at work were perceived positively. Overall, an expressed well team spirit and experienced social support was highlighted by the interviewees.


*“Apart from all the materials we have, the main support factor is the support of which is indeed being felt by leadership, colleagues, really, when meeting colleagues in the morning or is calling them, there is clearly a sense of unity. (…).” (Interviewee #7)*


#### 3.3.1. Social Support and Trust

Having the possibility to receive social support from colleagues when there is fear or overwhelming situations was also positively stressed by some of the outpatient caregivers. To trust their colleagues to carry out hygiene regulations correctly was another relevant resource to most of them.


*“I’m hoping for it, yes, I have to trust in that, otherwise we won’t be able to make it. I wouldn’t be able to work otherwise.” (Interviewee #6)*


#### 3.3.2. Communication

Communication has crystallized to be another important resource for outpatient caregivers, subdivided into communication at team as well as at leadership level. For one thing, there was communication with colleagues named specifically, which was rated very high by most of the outpatient caregivers. Apart from personal contacts, communication via telephone or digital mediums was used.


*“Yes, we are always exchanging information with colleagues, when something doesn’t seem right, they can then give advices and then one will calm down a little by talking to someone.” (Interviewee #5)*


A further aspect illuminated was the communication superior-side. A few answers let assume that communication with superiors might be improvable, precisely because there was a higher need in communication since the beginning of the pandemic for some of the interviewees. However, most of the outpatient caregivers felt like they could communicate with superiors on an open basis. For the most part, superiors had gotten a positive reputation for being available and helpful in problematic cases. Ways to achieve a conversation with them were either personal, by telephone or by making use of digital media. 


*“When there is worry or fear my superior has always got a ready ear. For instance, I’ve called her on the weekend. Yes, it was the weekend and she was not in the office but I’ve called her anyways. She even was afraid of and worried for the virus, too as well as for the patients and employees. (…). It helped me a lot, she was positive with me and through that I’ve found strength. She is no psychologist but my boss of course. She couldn’t do more.” (Interviewee #10)*


#### 3.3.3. Exchange of Information

An exchange of information about the COVID-19 pandemic took place in personal conversations, either in teams or by individual chats. In many cases, digital distribution lists were used, sometimes there were also notice boards or flyers mentioned. The majority found the exchange for information as sufficient and on a regular basis. 


*“Thus concrete, most of the times we get the latest information and hygiene regulations which are currently valid and we get many facts to bring information over to the patients and they will get educated. And we also had bigger tables about/ we really had conversations about the coronavirus and about fear and so on. We already had two meetings to speak about that with each other.” (Interviewee #6)*


#### 3.3.4. Appreciation

One more underlined resource in times of the pandemic was the feeling of appreciation. There was, on one hand, the appreciation by superiors and/ or colleagues pointed out by some few interviewees. On the other hand, there was appreciation by fellow human beings, for example patients and neighbors. They reported being appreciated in general for pursuing their profession as an outpatient caregiver. However, many interviewed persons have felt like they were being more appreciated since the start of the pandemic. Only a few complained about no or low appreciation by society, patients or surroundings in general.


*“Well yes, they are very proud of us that we keep working despite of the corona pandemic and that we keep being there for them. (…). Exactly. And our care service manager as well and other people, too.” (Interviewee #5)*



*“(…) the building I’m living in, many older people and one family are living there, and somehow at the beginning of my moving in, no one really knew that I’m a geriatric nurse and now they know they find it very, very nice. And so I’m clearly noticing appreciation and especially in these times, where not everybody was forced to leave the house and many were given the possibility of home office, everyone noticed: “Okay, he nevertheless leaves the house at 5 am to get to work”. I feel appreciation by seeing that people are joyful and find it nice to see what I’m doing. Yes, I’ve been told that many more times in the current situation.” (Interviewee #7)*


### 3.4. Strain Reactions Experienced by Outpatient Caregivers 

The following strain reactions were perceived by outpatient caregivers due to their job demands while working during the COVID-19 pandemic.

#### 3.4.1. Depressive Symptoms

Some outpatient caregivers have stated that demanding factors have resulted in several strain reactions. Sadness of outpatient caregivers have been caused by the lack of social contacts as well as restrictions of leisure activities lately. Other strain reactions such as listlessness as well as tiredness were mentioned frequently by surveyed outpatient caregivers. 


*“Listless. You can only go to work, do homework with your child and then you go to sleep. And her school’s homework are enormous, what they expect. Well, what does depressive mean, well, yes, I would say it like that.” (Interviewee #1)*


#### 3.4.2. Anxiety

In daily life, outpatient caregivers have also perceived anxiety occurrences, which sometimes happened in the outside world for instance while grocery shopping in the supermarket as well as while using public transportation. Less frequently, fears about death, general fear of the coronavirus and its duration were described.


*“Well, especially at the beginning, by the time everyone was allowed to work from home and one had the feeling to be cannon fodder. Oneself was technically much unprotected at the beginning because until then there was no / it wasn’t like that that everybody was working with a mask.” (Interviewee #11)*


#### 3.4.3. Fear of Consequences of the Pandemic

Many outpatient caregivers rather described their fear of possible consequences caused by the coronavirus pandemic. In many cases, the fear of the economic effects globally was highlighted, but also the fear of the own insolvency. Moreover, being afraid of fatal consequences and being scared for vulnerable people groups were also reported. A social distance in general between human beings was also something they worried about. Looking at the aspect of future, outpatient caregivers were concerned about their children’s future lives. Most; however, were afraid of the unknown and the uncertainty about the whole situation.


*“And in this future with a lot of this virus, we all are scared because we don’t know, what will come the next day? That is the uncertainty. (…) What will come afterwards? What’s the next step?” (Interviewee #10)*


#### 3.4.4. Individual Daily Performance

While a few outpatient caregivers haven’t experienced any kind of impact on their daily performance or even an increased one, most of the outpatient caregivers told about an impaired individual performance in their daily lives in contrast. Reasons were, above all, the double burden through caring for their children (e.g., home-schooling in their private lives as well as the additional hygiene measures they had to carry out since the start of the coronavirus pandemic). Private conflicts, having to wear masks all the time and a general tiredness also came up during the interviews. 


*“Yes, well, that is indeed a lot. Because of the fact that children are at home and a full-time job on top, and then doing home-schooling with three kids, that too is a task. Apart from my job, it is a challenge.” (Interviewee #8)*


#### 3.4.5. Perception of Stress

On account of all these factors, many outpatient caregivers have felt a general higher perception of stress. Different reasons were named in this phase of the interviews. However, mostly outpatient caregivers’ felt stress perception was favored by the pandemic situation overall. Private conflicts as well as the described double burden of home-schooling their children after work were also responsible for outpatient caregivers perceiving a greater feeling of stress. Stricter hygiene rules, mask obligation, a lack of PPE and social contacts, and examinations by the medical service of the health insurance companies or more work in general stressed outpatient caregivers, too.


*“Yes, I’m more irritated. I am indeed more irritated, I notice that at home I have never griped at my daughter that much.” (Interviewee #1)*



*“Yes, with the/ the self-incrimination for self-protection is higher, definitely the perception of stress.” (Interviewee #2)*


In adherence to the JD-R Model by Bakker and Demerouti [14], the results are visualized in Figure 1. Several job demands in work organization and task, quantitative and emotional demands and social relationships promote developing negative strain reactions such as stress or depressive symptoms. Job resources (e.g., good communication, social support and security) can buffer these job demands and have a positive effect on outpatient caregivers’ motivation to care during the COVID-19 pandemic (cf. [14]).

## 4. Discussion

### 4.1. Discussion of Interview Results

This is the first study applying qualitative methods to comprehensively examine the job demands, job resources as well as resultant strains experienced by outpatient caregivers in Germany in the course of the COVID-19 pandemic. Important insights into the working conditions associated with the COVID-19 pandemic of outpatient caregivers from Northern Germany were gained. Outpatient caregivers faced new and specific kinds of demands during their work, especially due to new hygiene and governmental regulations, causing, for instance, higher quantitative demands. Having to wear masks permanently during work was described frequently as the most demanding factor since the outbreak of the coronavirus. In this context, the lack of PPE was also mentioned by many interviewees. Simultaneously, stricter hygiene regulations were required constantly. Interviewees experienced new emotional demands while working in the outpatient care during the COVID-19 pandemic. Outpatient caregivers who were parents had a double burden (e.g., childcare after work) and a higher perception of stress in general. Key resources related to their profession were social interactions with clients, the communication with superiors and colleagues and the team spirit felt in the care services. Strain reactions were not only depressive symptoms, like anxiety or listlessness, but also the perception of stress as well as the fear of the pandemic’s future consequences. We generally found the proposed differentiation between job demands, job resources and strains in adherence to Bakker and Demerouti [14].

#### 4.1.1. Job Demands Faced by Outpatient Caregivers during the COVID-19 Pandemic

In this study, a higher working time was a demanding factor regarding the work organization for many of the outpatient caregivers, who stressed that filling in for staff who were sick or in quarantine increased the problem. A study among German semi-residential and outpatient care facilities also identified higher employee absenteeism as well as an additional workload (on average 40 min per shift) due to the pandemic and the resulting changes in working conditions [49]. Similarly, studies conducted among nursing home or hospital personnel reflected higher levels of workload [63] and increased work hours [64] during the COVID-19 pandemic. Caregivers in the present study experienced new duties (e.g., concerning hygiene and fever measurements) as demanding and increasing their quantitative demands. The additional effort required to implement the necessary protective measures was also mentioned in a German survey of semi-residential and outpatient care facilities, although only about one third of the participating facilities in that survey reported to conduct clinical monitoring of clients on COVID-19 (e.g., measurement of body temperature) [49].

With regard to their work tasks, a job demand that arose in the present study was the possible handling of coronavirus-positive patients. Even though none of the interviewees had a positive case among their patients, this was an issue that bothered some of them. In an online-survey conducted in the period April–May 2020 of semi-residential facilities and nursing services in Germany, about 30% (189 of 627) of the participating nursing services had been affected by confirmed or suspected cases of COVID-19 among clients [49]. It is important that caregivers know exactly what to do in such cases and can generally follow clear guidelines and pandemic plans [65]. Only a few participants in the present study mentioned having an in-house pandemic plan in this context. The World Health Organization (WHO) [66] published an interim guidance for home care for patients who might present coronavirus symptoms. In Germany, the Robert Koch-Institut [67] provides a checklist for the preparation of a company pandemic plan. Furthermore, interviewees of the present study emphasized feelings of high responsibility and self-reliance with regard to their work tasks, which has also been highlighted elsewhere [39].

In the context of their work environment and tools, interviewees perceived wearing masks, thus having trouble breathing, as very stressful, which was also mentioned by healthcare workers in hospitals [64]. However, it is strongly recommended that outpatient caregivers wear mouth-nose protection when working with patients to minimize the risk of infection for those risk groups [4]. Likewise, many of the participating outpatient caregivers in this study experienced a lack of PPE as burdening. This limited access to or shortage of PPE was described by several studies on healthcare workers in primary care, hospitals, nursing homes or outpatient/community-based settings (e.g., in the USA, Spain, Latin America and Germany), especially at the beginning of the pandemic (e.g., [49,63,68,69,70,71,72]). Furthermore, the lack of PPE was significantly associated with a higher fear of contagion [63]. U.S. nurses who reported receiving inadequate PPE were more likely to report symptoms of depression, anxiety and post-traumatic stress disorder (PTSD) [68]. 

Another demand for caregivers was to handle the fears and worries of their patients and also destructive or aggressive behaviors. Experiences of verbal or physical aggression is widespread among workers in the care sector in Germany [73,74]. This problem can be aggravated by the fact that the pandemic is also a burden for people in need of care (e.g., due to social isolation) [75] and that communication is more difficult (e.g., by wearing masks facial expressions cannot be recognized by others). In this context, the interaction with relatives is also important. A Germany-wide survey revealed that 25% of 1000 caring relatives reported that they could rather not well or not well at all explain the corona situation to the person in need of care or calm them down [76], which emphasizes relatives’ need for support.

The interviewees were confronted with emotional demands due to the pandemic such as experiencing patients’ sadness, fear and loneliness. The emotional impact of contact with death and suffering, including negative feelings because patients do not get family visits, was addressed before among nursing home professionals. Greater exposure to suffering and death was one factor explaining much of the variance in secondary traumatic stress scores [63]. Another job demand mentioned for caregivers in the present study was experiencing discrimination by patients, relatives and the public. Former studies stressed stigmatization by the community in the form of fear, disgust and discrimination of hospital and geriatric care workers [64,77,78]. Stigma positively predicted burnout and fatigue and negatively predicted satisfaction among frontline healthcare workers [77]. Discrimination by patients and relatives could be a specific additional demand for outpatient caregivers as they work in the homes of these people and are; therefore, exposed, for example, to their individual hygiene requests in contrast to hospital workers (cf. [21]). Another important demanding factor for outpatient caregivers was the fear of getting infected themselves or infecting others such as patients or family members, which was described in several studies before by healthcare workers and managers of care facilities [64,71,79]. In this regard, the wish of employees for sufficient health protection and the provision of PPE [71] is to be taken very seriously to reduce this burden on healthcare workers.

On the personal level, outpatient caregivers suffered from social isolation due to social distancing, which can be attributed to the imposed restriction of contact by the German government on 15 April to curb the COVID-19 pandemic [80] and was mentioned by former studies as well [64,71]. A changed social dynamic and social distance can be very demanding in these times, where humans need the most social support [81]. Rheindorf et al. [71] further stressed a lack of work balance due to social restrictions in this context. More virtual and social-distanced support were reported by other healthcare workers [64], but these options were perceived as less satisfying by interviewees in the present study. Furthermore, interviewees experienced work-family conflicts while having to work and to take over care of their children or even home-schooling. This was not specific to the German context, but effected, for example, also healthcare workers in the U.S. due to closure of childcare and schools in the course of the pandemic [64].

#### 4.1.2. Work-Related Resources

In adherence to the JD-R Model [14], job resources can have a buffering effect on job demands which can lead to milder resultant strain reactions [15]. Former studies on outpatient caregivers have shown that meaningfulness, social support and information or reflection in the team are seen as important job resources for this occupational group [21,29,73]. For example, working in an active team is already known in the pre-corona context to decrease stress and strain factors as well as make employees more confident in their work [21]. According to the results of this present study, these findings have not changed during the COVID-19 pandemic. In fact, interviewees stressed that social support and trust among colleagues and with superiors were even more important in this specific new situation outpatient caregivers found themselves in. Previous studies on healthcare professionals working in hospitals during the pandemic also identified team work, social support and emotional support as important job resources [64,82]. In an earlier study on clinic personnel (>50% nursing staff), social support was also seen as a resource as it was associated with higher self-efficacy during the COVID-19 pandemic [38]. Additionally, social support has been scientifically proven to be a mediator of mental strain factors among healthcare workers of a hospital in China c.f. [83]. Another study by Blanco-Donoso et al. [63], analyzing Spanish nursing home workers’ situation during the COVID-19 pandemic, makes clear that supervisor and co-worker support rank among job resources at a high level, as also mentioned by our interviewed outpatient caregivers. Working under respectful superiors was further described as a positive factor during the pandemic by interviewees. According to Zhao et al. [84], inclusive leadership could be a mediating factor regarding distress and important for nurses in hospitals during the COVID-19 pandemic. Feeling more appreciated for their work at their workplace and/or by fellow people was another relevant job resource of our interviewees. This was also mentioned by former studies on healthcare professionals working in hospitals during the pandemic [64,82]. 

Considering the fact that Germany called a social distancing plan by law to curb the pandemic [80], personal conversations and meetings might be missing even more than before in outpatient geriatric care. In this context, our interview results still highlight a sufficient communication with colleagues and/or superiors although there were no team meetings anymore due to the COVID-19 pandemic. Some of the outpatient caregivers have further brought up their satisfaction with their workplace regarding the COVID-19 pandemic generally, for instance in terms of available PPE, employer support as well as feeling secure concerning hygiene regulations. This was also shown in a review on healthcare workers, besides acceptance of the risk of infection and training for pandemic situations [13].

Hennein and Lowe [64] identified telehealth as another job resource for healthcare professionals in hospitals. Telehealth was not mentioned by interviewees in the present study. This can be explained by the fact that technology such as telehealth is not often applied in outpatient geriatric care so far in Germany. Although, potential is seen by nursing staff in areas such as video prompting to take medication or to go to the toilet, transmission of vital signs via video and providing services of everyday accompaniment and nursing care via video [85].

#### 4.1.3. Resultant Strain Reactions Experienced by Outpatient Caregivers

Interviewees reported several strain reactions caused by the job demands during the COVID-19 pandemic. Our study results show that depressive symptoms (e.g., sadness, listlessness and tiredness) are distinctively perceived by outpatient caregivers. Studies on nurses from the stationary setting [13,30,31,35,36,42,64,68,86] show that depressive symptoms seem to be common strain reactions to work-related demands since the start of the outbreak of the coronavirus. A further strain reaction mentioned by our interviewees was anxiety due to the COVID-19 pandemic experienced in everyday working life. Taking a look at the inpatient care, the pandemic has also caused anxiety in nursing staff generally [30,32,35,36,38,42,68,86]. Fears of own or family’s infection were mentioned by our interviewed outpatient caregivers in the context of job demands during the coronavirus pandemic and were also factors that led to feelings of anxiety in the hospital setting [39,46,64,86]. Lack of PPE was a job demand stressed by our participants which resulted in fears in the stationary setting [45,46,70]. A less frequent reason of anxiety by our respondents was the fear of death due to the coronavirus, also found by Zhang et al. [46]. Fears of the COVID-19 pandemic’s consequences were especially described by our interviewees in terms of financial difficulties which was also found by other researchers [47,49]. Individual daily performance of respondent outpatient caregivers was partially impaired (e.g., because of a double-burden at home, private conflicts or daily mask obligation during work). This has not been come up in the scientific literature so far. However, impairment of mental health in general was found in other studies focusing on the inpatient care [33,46,87] or even physical strains like headache [71,86], musculoskeletal diseases [71], throat pain [86], troubled breathing [46] or impairment of daily performance in general were discovered [71]. A higher perception of stress was described by surveyed outpatient caregivers due to the COVID-19 pandemic and its following stricter hygiene regulations. Studies on the inpatient care sector also reported a generally increased stress perception due to the pandemic [13,30,37,40,63,70,86,88] or even burnout symptoms [77], which can be caused by a continuous perception of stress [89]. Moreover, other studies showed further strain reactions experienced by nursing staff of the stationary setting caused by the coronavirus pandemic such as worrying/nervousness [40,41,45], discomfort and helplessness [39], exhaustion [46,47], insomnia [30,86], fatigue [39,77] and PTSD [13,32,34,86]. PTSD was also found among outpatient caregivers [68], as well as feeling threatened [45] or even having suicidal thoughts [40,41]. A positive strain reaction—decrease of stress perception due to work resources—was shown in one study on nursing staff [84].

Some studies specifically focused on psychological strain reactions in the course of the COVID-19 pandemic of frontline nurses who are directly involved in the treatment of COVID-19 patients [30,34,41,43,46,63,68,88] and observed strain reactions such as depressive symptoms, anxiety and fear [30,39,41,43,46,86] as well as increased perception of stress [30,43,63,88] and insomnia [30,86], fatigue [39,43] or traumas [34]. Furthermore, physical symptoms like headache, throat pain, anxiety and lethargy were experienced by frontline nurses [86]. Fear of infection and infecting others were also described in a few publications about nurses treating COVID-19 patients [46,86].

All in all, it should be noted that most of the studies discussed at this point are based on the inpatient care sector and might; therefore, report other and more severe perceived strain reactions since participants in these studies were sometimes directly involved in the treatment of COVID-19 patients. Finally, it should be considered that our study included only three male participants so the majority of the outpatient caregivers was female. Women in the care profession seem to perceive negative strain reactions on higher levels during the COVID-19 pandemic. This was shown in former studies for mental disorders in general [33], anxiety and post-traumatic stress [32], symptoms of depression, anxiety, insomnia and distress [30], feeling threatened by the pandemic [45] or compassion fatigue and burnout syndromes [77], which were especially perceived by female participants.

### 4.2. Strengths and Limitations

A strength of our study is the fact that we have recruited outpatient caregivers from different care services (of different city districts of Hamburg, Germany) with varying sociodemographic characteristics (e.g., different ages and length of work experience) in a very small time span as a quick reaction to the sudden coronavirus outbreak. Therefore, we were able to establish a broad picture of the COVID-19 pandemic situation and its specific demanding and resource factors for outpatient caregivers from Germany, which have not been examined yet. To increase the trustworthiness of our findings, we employed rich descriptions of our results and displayed many direct quotes from the interviewees [90]. Furthermore, research results were profoundly discussed in a group of researchers and were also contrasted with empirical references. 

However, it should be noted that our findings are based on a random sample which was partly achieved via a snowballing technique, which might have increased the risk of self-selection among the participants. For instance, people having interest in the topic in the first place are more likely to participate. Moreover, only three of our interviewees were male. However, women tend to participate in studies more often than men [56] and more women than men are employed in care services in Germany [91]. 

Further methodological limitations are that we conducted telephone interviews instead of face-to-face interviews. Telephone interviews are still based on the relationship between interviewer and interviewee but on the one hand eye contact was not given and on the other hand there might have been a distanced conversation atmosphere [92]. Additionally, there was an asynchronous communication due to the telephone implied as well as a decrease in possible social clues [93,94]. 

Another limitation of our study may be seen in the relatively small sample size so that results must be reviewed in terms of transferability and generalization [52,55]. All in all, generalizations of our results are impeded by the nature of our qualitative research design. However, individual interviews can give significant statements [55] and data saturation seemed to be achieved. This is likely to occur within the first twelve interviews [95]. Nevertheless, the results of the present qualitative study should be verified by studies with larger samples and especially by quantitative studies that could provide broader knowledge on the topic.

### 4.3. Implications for Further Research and Practice

Further research studies with larger sample sizes are needed as outpatient caregivers depict a special group of employees who in contrast to others are forced to go out and visit their patients no matter the distance regulations. Thus, job demands might be higher in general for them during this current pandemic [96]. In such studies, it would be interesting to conduct interviews with outpatient caregivers who have experienced the coronavirus pandemic while it lasts and how their perceptions of job demands and resources due to the pandemic might have changed over the time. Especially given the fact that this is a whole new exceptional situation for all human beings simultaneously. Finally, it could be a future interest in research to not only expand the sample size but also to achieve a more representative study sample (e.g., characteristics which should be considered could be different ages and an even gender distribution, using a quantitative questionnaire study). Part-time and full-time workers could also be differentiated by analyzing and comparing their job demands, resources and strain reactions. Apart from this, outpatient caregivers and their demands and resources during the coronavirus pandemic should be researched worldwide, especially in countries which have high mortality rates due to COVID-19. Eventually, it would be of scientific interest to enlighten the perspective of outpatient caregivers’ job resources during the coronavirus pandemic as there are barely publications addressing them, although they do have a buffering effect on the experienced job demands according to Demerouti and Nachreiner [15]. After more research has been carried out, specific interventions within the framework of work-side health promotion and occupational health and safety could be developed and implemented as specific job demands could be addressed. 

Implications for further practice could be divided in behavioral (improving coping competences) and structural prevention (changes in the work organization and environment) [97,98,99]. On the behavioral level, it might be sensible to strengthen individual caregiver’s personal resources concerning their ability to protect themselves in terms of hygiene. Furthermore, in order to enhance outpatient caregivers’ resilience, trainings to strengthen personal resources are recommended (cf. [99]). On the structural level, there could be several things done to ensure safe working. Besides educating their employees with regard to infection control on a regular basis, employers could also try to enlighten patients and their relatives themselves directly. Topics would be how patients’ safety is ensured and which actions are taken to keep risk of infection to a minimum. Patients’/relatives’ worries could be decreased in that way, which might have a demand-reducing effect to outpatient caregivers. PPE and other relevant working tools such as disinfectants, gloves and fever thermometers, should be provided at all times so any shortages of materials would be excluded in case of future possible emergencies due to an outbreak. To supply masks and disinfectants for patients’ homes could also be reasonable to secure both sides. To comfort outpatient caregivers it could be thought of governmental regulations to make rapid antigen tests available for outpatient caregivers in case there is a suspicion of a coronavirus infection [100]. Given the positive effects of social support and cultivated communication [101], team spirit, communication and social exchange between colleagues and superiors should be ensured despite of the actual pandemic situation [81,102]. Moreover, the use of digital tools could relocate office work to specific time spans where outpatient caregivers would be able to work from home so working time outside from home could be decreased which could be advantageous during the pandemic. With the objective to relieve outpatient caregivers from their double-burden of home-schooling duties after work, employers could offer possibilities of childcare at the workplace as well as flexible working times. By doing so, individual resources of outpatient caregivers could be saved and resorted elsewhere (e.g., at work).

All-in-all, the experiences from the COVID-19 pandemic should be used in order to ensure a better preparation for emergency scenarios in the future where it also could be reasonable for nursing care funds and public health departments to work together (cf. [103]).

## 5. Conclusions

The present study focused on specific job demands, resources and their subjectively perceived effects of outpatient caregivers during the COVID-19 pandemic, a yet unexplored field. Respondents in our study reported a variety of specific job demands in their working activity due to the COVID-19 pandemic which they have to deal with every day. Despite described burdens, various work-related resources were expressed by respondents. Besides, resulting negative strain reactions due to specific job demands experienced by outpatient caregivers illustrate the relevance of reducing pandemic-related job demands. Despite the limited number of interviews conducted, our results suggest that, in the future, specific occupational health and safety measures have to be conducted to be prepared and to ensure employees health and safety in the outpatient care, especially in sudden emergencies such as the current coronavirus pandemic. Finally, our results provide an adequate basis for developing specific health promotion measures for a sudden outbreak of infectious diseases.

## Figures and Tables

**Figure 1 ijerph-18-03684-f001:**
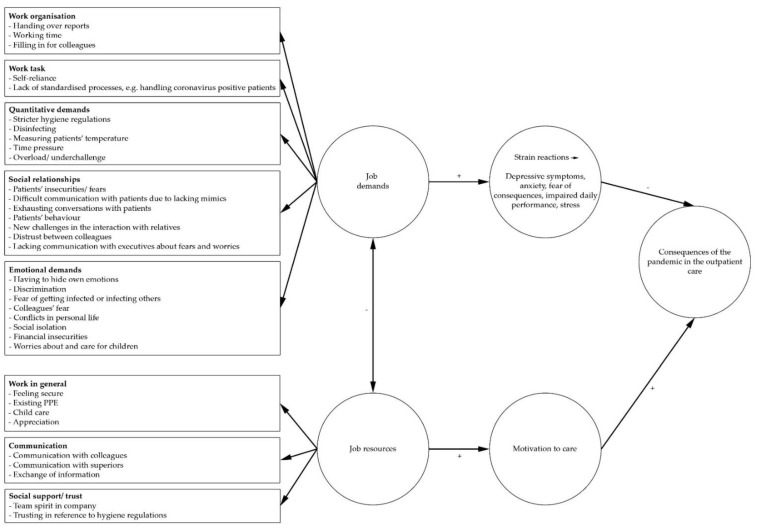
The Job Demands-Resources model, own depiction (following [16]).

**Table 1 ijerph-18-03684-t001:** Interview topic list.

Phase of the Interview	Contents
1 Information phase	Introduction: Study information, confidentiality, informed consent
2 Warm-up phase	Qualifications, working activity
3 Main phase	Work-related job demands, emotional demands, effects on personal life Resources in work life Strain reactions
4 Final phase and end of the interview	Socio-demographics of the interviewees and farewell

**Table 2 ijerph-18-03684-t002:** Participant characteristics.

ID	Gender ^1^	Age	Children in Household	Date of Interview (MM/DD/YY)	Qualification	Occupation	Work Experience as an Outpatient Caregiver	Work Schedule
1	f	31	1	05/07/2020	Caregiver	Outpatient geriatric nurse	7 months	Full-time
2	f	31	0	05/07/2020	Geriatric nurse	Outpatient geriatric nurse	9 years	Full-time
3	f	33	1	05/07/2020	Geriatric nurse	Outpatient geriatric nurse	7 months	Full-time
4	m	64	0	05/08/2020	Geriatric nurse	Outpatient geriatric nurse	36 years	Full-time
5	f	21	0	05/12/2020	Home and family care	Outpatient home and family caregiver	1 year	Full-time
6	m	51	0	05/12/2020	Geriatric nurse	Outpatient geriatric nurse	5 years	Full-time
7	m	25	1	05/15/2020	Geriatric nurse	Outpatient geriatric nurse	1.5 years	Full-time
8	f	38	3	05/15/2020	Healthcare and nursing staff	Outpatient caregiver	16 years	Full-time
9	f	51	0	05/19/2020	Geriatric nurse	Outpatient geriatric nurse and office manager in health sector	23 years	Full-time
10	f	36	0	06/03/2020	Social manager	Outpatient caregiver	4 years	Full-time
11	f	46	1	06/11/2020	Geriatric nurse, additional qualification intensive and palliative care	Outpatient geriatric nurse	20 years	Full-time
12	f	50	2	06/11/2020	Wound expert	Care specialist and nutrition manager in the outpatient care	10 years	Full-time
13	f	34	0	06/15/2020	Geriatric nurse	Care specialist and deputy care management in the outpatient care	5 years	Full-time
14	f	67	0	06/19/2020	Geriatric nurse	Outpatient geriatric nurse	24 years	Part-time
15	f	37	0	06/29/2020	Geriatric nurse and wound expert	Outpatient geriatric nurse and wound expert	5.5 years	Part-time

^1^*n* = 15; f = female, m = male.

## Data Availability

The data analyzed during the current study are not publicly available due to German national data protection regulation. They are available on individual request from the corresponding author.

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
