# Peer review of "Job Demands, Resources and Strains of Outpatient Caregivers during the COVID-19 Pandemic in Germany: A Qualitative Study"

_ijerph, 2021, doi:10.3390/ijerph18073684_

Round 1

Reviewer 1 Report

The  paper shows an original and important study about outpatient caregivers.  The was is to describe how they perceive their working conditions during the pandemic in Germany and the difficulties they face. 
It is a qualitative study

The  paper shows an original and important study about outpatient caregivers.  The aim is to examine how they perceive their working conditions during the pandemic in Germany and the difficulties they face. 
It is a qualitative study with telephone interview.

 The use of The Job Demands-Resources Model (JD-R Model) by Bakker and Demerouti gives a high degree of consistency to the results obtained. I think it is very suitable.
It would be interesting for the authors to go deeper into the conceptual framework of this model in order to understand the results obtained. 
The results are clearly presented and support the conclusions of the study.

Author Response

Thank you very much for your feedback. We have further explained the theoretical model (JD-R Model) on page 2. Moreover, we have further shed light on our created Figure 1 by highlighting the link between our results and the model (page 18/19).

Reviewer 2 Report

Dear authors,

The manuscript deals with the current theme, but some questions are asked due to the need to improve the theoretical discussion and possible selection bias.

Introduction
it is necessary to better define who are the outpatient caregivers (professional categories) involved in this context

Theoric model
How does the theoretical model vary between the different outpatient caregivers included?

Method
Knowing which professionals are active, why not choose just one professional category?
Does the group of researchers have an influence on the predominance of participating nurses?

Limitations:
How does the “mixture of categories” or the predominance of one among all contribute or limit the study?

Author Response

Dear authors,

The manuscript deals with the current theme, but some questions are asked due to the need to improve the theoretical discussion and possible selection bias.

Introduction
it is necessary to better define who are the outpatient caregivers (professional categories) involved in this context

Thank you very much for your feedback. In the introduction we have added the most common professions of outpatient caregivers in Germany and we have explained their job activities for further understanding.

Theoric model
How does the theoretical model vary between the different outpatient caregivers included?

The Job Demands-Resources Model (JD-R Model) by Bakker & Demerouti is a universally applicable model that considers occupation-specific or rather activity-specific demands and resources. An imbalance of job resources and job demands can lead to negative strain reactions in the long run. Our study primarily examines the work activity as an outpatient caregiver, which, as also added in the introduction, does not differ in essence, regardless of which profession in the outpatient care is being practiced.

Method
Knowing which professionals are active, why not choose just one professional category?
Does the group of researchers have an influence on the predominance of participating nurses?

In the introduction we have pointed out that there are several occupations in the outpatient care by carrying out the same job activities. Therefore we wanted to examine the superordinate employment group of outpatient caregivers in Germany. For this reason, we did not set any specifications for the respective profession. However, we added a further explanation in the methods part (page 5). The fact that most respondents are geriatric nurses is also reflected in the statistics (see supplement in the introduction), but was not a prerequisite for the research group. Participants were not known personally until the time of the interviews and there was no influence on the predominance of participants.

Limitations:
How does the “mixture of categories” or the predominance of one among all contribute or limit the study?

We were able to recruit 15 outpatient caregivers working in Hamburg in different occupations in outpatient care. Perceived job demands, resources and strain reactions could be reflected to the COVID-19 pandemic situation as all of them were already working in the outpatient care before the outbreak, so comparisons or rather specific perceptions of the individuals were possible. In addition, it was our aim to examine pandemic-specific demands, resources and strain reactions of the superordinate employee group of outpatient caregivers. Occupations in the outpatient care in Germany are widespread in geriatric, health and home care, whereby their job activities are quite the same (see introduction, page 2). Therefore we did not expect any influence on our results by not presupposing a specific occupation.

Reviewer 3 Report

I found the article interesting in terms of content, and it has great potential in terms of academic interest. But with a sample of only 15 subjects, it is impossible to achieve the objectives proposed by the study. I suggest that the authors expand the study sample and resubmit the work.

Author Response

Thank you for stressing the academic interest in the topic of our study and also for your suggestion regarding our study sample. However, we do not believe that we have failed to meet the study objectives with our sample of n = 15 participants. We would like to kindly point out that we conducted a qualitative study. Qualitative research studies aim to provide a deep understanding and description of complex social phenomena from the perspective of the people under study (Creswell, 2013). In our study, we provide a complex insight into the job demands and resources as well as strain reactions of outpatient caregivers during the coronavirus pandemic in Germany. While quantitative studies seek to generate broad data from a large (representative) number of participants to reach statistical power, qualitative studies seek to generate rich data with thick (detailed and complex) descriptions from each participant (Clarke & Braun, 2013). Therefore, representativeness is not a useful criterion for qualitative samples (Helfferich, 2011). In addition, generalisability of the results is not a primary aim of qualitative research (Misoch, 2015). In order to determine the sample and assess generalisability in qualitative studies a narrow and precise definition of the group of interest, a wide variation within this group, and finally, narrowing of the group definition should be provided (Helfferich, 2011). In our study we included outpatient caregivers from outpatient care services in Hamburg, Germany working as an outpatient caregiver for at least six months, who were fluent in the German language. Within this group we had a wide variation of participants, e.g. in terms different care services, age and work experience to include different perspectives on the topic of job demands, resources and strains. In our limitation section we state, that only three of our participants were male (page 24). In addition, we have followed the principle of theoretical saturation (Helfferich, 2011; Guest, 2006), which means that we reached data saturation (likely after 12 interviews) and further interviews would not have led to the identification of essential new topics (see page 5 and 24). In qualitative research, this is actually more relevant than reaching a certain number of interviews.#

References:

Guest, G.; Bunce, A.; Johnson, L., How many interviews are enough? An experiment with data saturation and variability. Field Methods 2006, 18, 59-82.

Creswell, J. W. Qualitative inquiry and research design: choosing among five approaches. 3rd ed.; SAGE Publications: California, USA, 2013.

Helfferich, C., Die Qualität qualitativer Daten. Manual für die Durchführung qualitativer Interviews. 4 ed.; VS Verlag für Sozialwissenschaften: Wiesbaden, 2011.

Misoch, S., Qualitative Interviews. De Gruyter: Oldenburg, 2015.

Round 2

Reviewer 3 Report

Dear authors,
First of all, thank you very much for your prompt reply.

As you can conclude, I am not particularly enthusiastic about qualitative studies, which I understand as a starting point for knowledge about a reality, but they leave too many questions open. Could the results found be transferred to the COVID healthcare groups in other regions of Germany? Would the answers to the questions posed follow the same direction? And in other cultures, what would those answers be like? What are the benefits of knowing the application of the model to such a small group. However, I found it pleasant to read and the study objectives adequate.

Having said all this, and although I do not agree on the way in which to respond to the very interesting proposals of interest raised by the authors in the manuscript, if the editorial committee of the journal and my review colleagues accept the publication I I also give my approval. Of course, I would ask you to please include in future lines of research the need to include quantitative analyzes alongside qualitative ones in order to obtain a broader view of the knowledge about the main objective of your work.

I reiterate my thanks to both the publisher and the authors for how simple and fast the review process of this manuscript is.

Author Response

Thank you very much again for your feedback. We included a further sentence on the need of studies with larger samples and especially quantitative studies for further verification of our study results at the end of the limitation section on page 25.   

Furthermore, we reviewed the entire manuscript for English language and style and made some final minor corrections.